# Presenting decision-relevant numerical information to Dutch women aged 50–70 with varying levels of health literacy: Case example of adjuvant systemic therapy for breast cancer

**Inge S. van Strien-Knippenberg**[1]\*, **Daniëlle R. M. Timmermans**[1], **Ellen G. Engelhardt**[2], **Inge R. H. M Konings**[3], **Olga C. Damman**[1]

**1** Department of Public and Occupational Health, Amsterdam Public Health Research Institute, Amsterdam University Medical Center, Vrije Universiteit Amsterdam, Amsterdam, the Netherlands, **2** Division of Molecular Pathology, The Netherlands Cancer Institute-Antoni van Leeuwenhoek Hospital, Amsterdam, the Netherlands, **3** Department of Medical Oncology, Amsterdam University Medical Center, Cancer Center Amsterdam, Vrije Universiteit Amsterdam, Amsterdam, the Netherlands

\* i.vanstrien@amsterdamumc.nl

## Abstract

### Background

If communicated adequately, numerical decision-relevant information can support informed and shared decision making. Visual formats are recommended, but which format supports patients depending on their health literacy (HL) levels for specific decisions is unclear.

### Study aim

The aim of this study is to investigate: 1) the effect of survival rates and side-effects presentation formats on comprehension and 'feeling informed'; 2) differential effects among women with higher/lower HL, with adjuvant systemic breast cancer therapy as case example.

### Methods

Two online experiments among women from the Dutch population without a history of breast cancer were conducted. Experiment 1 had a 3 (survival rate format: text block–bar graph–icon array) x 2 (HL: low–high) between-subjects design. Experiment 2 had a 5 (side-effects format: no probability information–probability information in numbers with or without a visualisation–probability information in numbers with or without a visualisation accompanied by a description of the side-effects) x 2 (HL: low–high) design. Primary outcomes were comprehension and feeling informed (Experiment 2 only). Formats were previously designed in co-creation with patients.

**Data Availability Statement:** Data cannot be shared publicly because the Ethics Committee approved the collection and analysis of the data for the specific study only. The Ethics Committee requires that the data collected remains securely stored and not be shared publicly. Researchers who meet the criteria for access to confidential data can contact i.vanstrien@amsterdamumc.nl to request the data. In addition, the data can also be requested from the head of the Amsterdam UMCs Public and Occupational Health department via the departments secretariat Div10-POHsecretariaat@amsterdamumc.nl

**Funding:** This study was conducted within a collaboration project. The project was funded by the PPP Allowance made available by Health~Holland, Top Sector Life Sciences & Health, to stimulate public-private partnerships (Grant number LSH18079) and the Dutch health insurer CZ. The funding agreement ensured the authors' independence in designing the study, interpreting the data, writing, and publishing the report.

**Competing interests:** The authors have declared that no competing interests exist.

## Results

In Experiment 1, presentation format did not affect gist or verbatim comprehension. Higher HL was associated with higher gist comprehension. Experiment 2 showed an interaction between presentation format and HL on 'feeling informed'. When provided with visualised probability information without a description of the side-effects, women with lower HL felt better informed than women with higher HL.

## Conclusion

Visual formats did not enhance comprehension of survival rate information beyond a well-designed text block format. However, none of the formats could overcome HL differences. When designing decision-relevant information, visualisations might not necessarily provide an advantage over structured numerical information for both patients with lower and higher HL. However, a deeper understanding of presenting side-effect information is warranted.

## Introduction

Women with primary breast cancer face multiple decisions during their treatment trajectory, including concerning (neo-)adjuvant systemic therapy. Informing patients about benefits and harms of different options is one of the key principles in health communication regarding informed and shared decision making (SDM) [1, 2]. Benefits (e.g., prolonged survival, reduced recurrence risk) and harms (e.g., side-effects, lower quality of life) can be presented in the clinical encounter and patient decision aids (PtDAs) [3]. Personalised survival rates, i.e., survival rates obtained from prognostic models based on individual patient characteristics, are increasingly used in this respect [4]. Understanding information about the probability of experiencing harms and benefits of treatment options can help in the decision-making process. However, understanding medical probability information is greatly influenced by patient skills related to information processing, such as the ability to derive meaning from abstract information and factual evaluation and appraisal of information [5]. Such information processing skills are captured in concepts such as health literacy (HL), which is mainly about accessing, understanding, and using health information [6]; numeracy, which is about understanding and using numbers and probabilities; and graph literacy (GL), which concerns the ability to understand graphically presented information [7]. Concerning HL there is a variety of conceptualizations. While some conceptualizations focus on a broad and holistic view of HL that also includes, for example, searching for information or assessing the reliability of information [e.g., 8], other conceptualizations focus more on specific aspects of HL, such as the literacy aspect [e.g., 9]. In this study, the focus is on understanding and interpreting abstract health information. Therefore HL is conceptualized as an individual ability in which understanding and interpreting health information is central.

Best practices exist for presenting decision-relevant information to enhance patient understanding, such as numerical formats instead of verbal labels only [10]. Visual formats are recommended for certain information types, such as line graphs to show trends over time and bar graphs to compare multiple outcomes [11], although studies have not always compared such formats to a format without visualisation. Whether a visualisation has added value and which visualisation is best to use depends on the purpose of the communication [12] and the target group [13]. When using visualisations, it is important to adhere to design principles such as

clearly labelled captions and axes and consistent denominators/scales [4, 11]. Visualising numerical information can have benefits, mainly because less cognitive effort (e.g., fewer mental calculations) is required [14, 15]. Furthermore, patterns are more visible, comparing multiple options is easier [16, 17], and denominator neglect (i.e., tendency to pay more attention to numerators than denominators) can be reduced by conveying the part-to-whole relationship [18]. Reducing cognitive effort may be especially beneficial for people with lower HL or numeracy [11, 19], as they generally have more difficulty understanding decision-relevant information [11, 20, 21].

Information comprehension can be divided into gist comprehension, referring to the essential aspect of the information (e.g., comparing which quantity is greater) and verbatim comprehension which is the literal, detailed message content (e.g., how much larger a quantity is in exact numbers) [22, 23]. Several studies showed that visual formats improve both gist and verbatim understanding [11]. However, other studies found that visual formats are superior mainly for communicating gist messages and numerical formats for verbatim messages [4]. Besides improving understanding, visualisations can influence people's behavioural intentions and preferences [12]. Therefore, when designing a presentation format, it should be aligned to the communication goal and context [4, 12].

Although visual formats are generally mentioned in recommendations for probability communication, e.g., International Patient Decision Aid Standards (IPDAS) [4], due to variations in study design and outcome measures, there is no consensus on which format should be used in which situation [4, 24]. For example, a study about the presentation of benefits and harms of cancer screening showed no differences in understanding fact boxes with numbers and fact boxes with icon arrays [25]. However, the decision to undergo cancer screening versus cancer treatment is arguably different. In the context of adjuvant systemic breast cancer treatment, previous studies of visual formats of survival rates (i.e., icon arrays and bar graphs) have been conducted [26–28]. However, only one study compared icon arrays and bar graphs, showing a 2-option icon array was best understood compared to bar graphs and 4-option visualisations [27]. In none of the studies in the context of adjuvant systemic breast cancer treatment, a direct comparison with a non-visual format was made. Moreover, no side-effect information was presented, while this is essential in the trade-off to be made by patients. When studies presented information about side-effects of cancer treatments or treatments to reduce the cancer risk, the probability of side-effects was typically not presented visually [e.g., 29–32]. One study examined two visual side-effect formats, i.e., a bar graph and an icon array, and found no differences in understanding. However, no information about the benefits was presented and no comparison with a textual condition was made [33]. One study used a bar graph to display probabilities of side-effects in addition to text in the context of medication to reduce the risk of developing cancer; participants were found to be more accurate when viewing this bar graph compared to text alone [15]. However, the information on side-effects was limited and consisted of one side-effect only. This does not fully correspond to the variety of information on side-effects that is usually provided for cancer treatment such as adjuvant systematic treatment for breast cancer. In addition, previous research has not investigated which format best suits people depending on their information processing skills.

To gain more insight into how to present decision-relevant information in the context of systemic adjuvant treatment for breast cancer, we co-created several presentation formats regarding benefit (i.e., survival rates) and potential harm (i.e., side-effects) with patients with breast cancer in a previous study [34]. To account for potential HL differences, this prior study also included patients with low HL. Based on this co-creation study, bar graphs and icon arrays seemed promising for visualising survival rate information, also for those with lower HL. To compare these visualisations to non-visualised numerical information, a text block format was

developed. Regarding side-effect information, patients expressed a need for including probability information and explanations of what the side-effects exactly entail [34, 35], corresponding to previous research [36–38]. Therefore, five presentation formats were developed that varied in the way of presenting probabilities (no probabilities or probabilities in numbers/visualisation) and in providing an additional description of side-effects (for details see Methods section).

The current study aimed to investigate: 1) the effect of several presentation formats of survival rates on comprehension; 2) the effect of the provision of side-effects probability information and accompanying description of those side-effects on comprehension and feeling informed; and (3) differential effects of the presentation formats among women with lower HL versus higher HL. Since presentation formats of decision-relevant information can also influence patients' intentions and evaluations [12], we explored effects on several secondary outcomes: affect, hypothetical decision, decision confidence, and evaluation of information. In addition, we also assessed the perception of the treatment effect in the first experiment and risk perception regarding additional treatment in the second experiment. Because the information presented was intended to support decision-making, we also included decision uncertainty and the extent to which the information contributes to the perceived preparedness for decision-making in the second experiment. While some (elements) of the co-created visualisations were also tested in previous studies, our study adds the following. First, our information was similar to the complex information that can be provided in oncology practice, such as multiple side-effects with probability estimates with quite a large range. Previous studies simplified this information. Secondly, we also looked at the combination of information on survival rates and side-effects, because in SDM practice, both are needed to make a trade-off. Third, the formats were developed in co-creation with patients, so that the needs of the end-users were central during the development of the formats. Finally, we compared survival rates presented in icon arrays and bar graphs with a well-designed textual numerical format (Experiment 1), an underexposed comparison in previous studies.

## Methods

### Design and materials

In two online randomised experiments, women viewed presentation formats with hypothetical information embedded in an online survey. Experiment 2 was performed after the first and contained new participants. Participants in Experiment 1 were excluded from participating in Experiment 2 by the panel. Before data collection, we formulated hypotheses, shown in Table 1. Ethical approval was obtained from the medical research ethics committee of Amsterdam UMC, location VUmc (FWA00017598). The Dutch Medical Research Involving Human Subjects Act (WMO) did not apply. Participants provided written informed consent (online) after reading the study aim in the online survey. This study was pre-registered before data collection through the Open Science Framework on July 8[th], 2021 https://osf.io/sxpjf/?view_only=cb702fb0aa904758bc2ce4a19abf0b74. Deviations from the pre-registration are indicated.

The first experiment contained a 3 (survival rate format: text block–bar graph–icon array) x 2 (HL: low–high) between-subjects design. In the formats, three treatment options were presented: (1) no additional treatment; (2) hormone therapy; and (3) combination of hormone therapy/chemotherapy. The numerical information was an example of personalised information obtained from a prognostic model [39]. In co-creation with breast cancer patients, various survival rate visualisations were developed [34]. A bar graph and icon array seemed most appropriate in this context, although there were mixed results regarding the feelings evoked by icon arrays (i.e., some felt overwhelmed and others perceived them as more personal). To

**Table 1. Hypothesis of Experiment 1 and Experiment 2.**

| Experiment 1 | H1a | A visualisation of the survival rates (either a bar graph or an icon array) will lead to more adequate gist comprehension of the probability information compared to textual information only. |
|---|---|---|
| | H1b | A visualisation of the survival rates (either a bar graph or an icon array) will lead to more adequate verbatim comprehension of the probability information compared to textual information only. |
| | H1c | Those with lower HL will be better supported in comprehension with a visualisation (either a bar graph or an icon array)–compared to textual information–than those with higher HL. |
| Experiment 2 | H2a | Probability information about the side-effects will lead to more adequate gist comprehension of the trade-off, compared to no probability information. |
| | H2b | Visualised probability information about the side-effects will lead to more adequate gist comprehension of the probability of side-effects, compared to probability information in numbers. |
| | H2c | Those with lower HL will be better supported in comprehension with the visualised probability information–compared to probability information in numbers–than those with higher HL. |
| | H3 | Probability information about the side-effects and an accompanying description of the side-effects will lead to feeling more informed, compared to no information. HL might influence this effect. |

compare visualised data with textual numerical data, a text block format was developed that did not visualise the numerical information but did use visual elements (e.g., three text blocks to indicate three options). Fig 1 displays the survival rate formats used in Experiment 1.

We built on the best-understood survival rate format from Experiment 1 in Experiment 2. This survival rate format would be the same for all participants in Experiment 2. This second experiment contained a 5 (side-effects format: no probability information–probability information in numbers without description–visualised probability information without description–probability information in numbers with accompanying description–visualised probability information with accompanying description) x 2 (HL: low–high) between-subjects design. The five side-effects presentation formats are described in Table 2 and S1 File. Fig 2 shows the format with the most extensive information. The first format contained no probability information, which resembles how side-effect information is often presented to patients. The other formats contained probability information in numbers or a visualisation. Additionally, some formats contained contextual information (whether a side-effect disappears after treatment and whether something can be done about this side-effect). This need for contextual information was expressed by patients in the preceding co-creation sessions [34]. The fact that we used likelihood estimates in a fairly wide range was driven by clinical reality. We wanted to investigate information formats that could be used in oncology practice and in the Netherlands (nor in most other countries, as far as we know) no exact point estimates were available. So, we developed a visualisation inspired by the results of the co-creation sessions (i.e., a horizontal bar graph), but also based on the available information (i.e., whether side-effects occur in 1–10 or more than 10 out of 100 women). The newly developed visualisation was pre-tested with eight women. An oncologist (I.R.H.M.K) reviewed the medical content of both experiments to ensure accuracy and compliance with practice in encounters with patients.

## Participants

In both experiments, we used a convenience sample of women from the general Dutch population aged 50–70 years without a history of breast cancer. In our preceding co-creation sessions, participants were (former) patients with breast cancer [34]. In the current study, we included women without breast cancer, since the decision-relevant information is intended for newly-

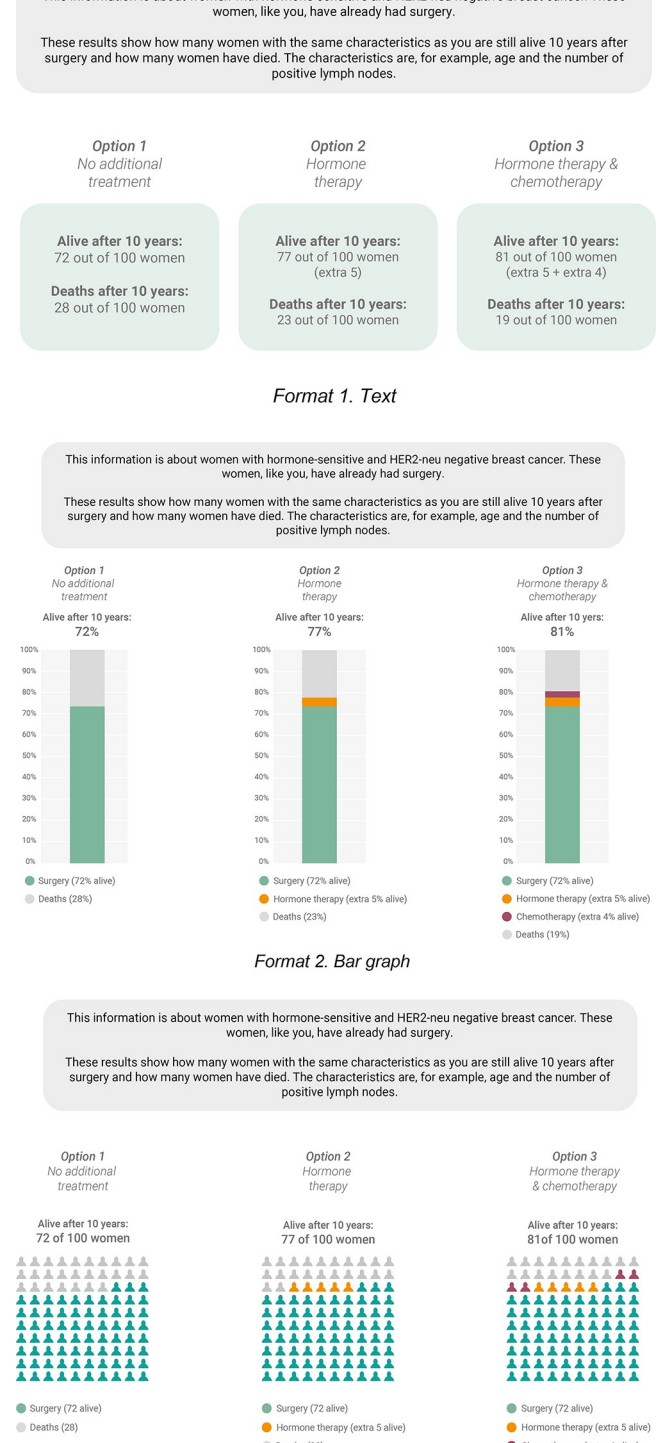

**Fig 1. The three survival rate formats.**

**Table 2. Description of the five side-effects formats of Experiment 2.**

| Format | Format description |
| --- | --- |
| A. No probability information | Side-effects are listed without numerical probability information and without additional information about the side-effects |
| B. Probability information in numbers without description | Side-effects are divided into two categories indicated by numbers:<br>• side-effects experienced by 1 to 10 out of 100 women<br>• side-effects experienced by more than 10 out of 100 women<br>• There is no additional information about the side-effects |
| C. Visualised probability information without description | Side-effects are divided into two categories indicated by visualisations:<br>• side-effects experienced by 1 to 10 out of 100 women<br>• side-effects experienced by more than 10 out of 100 women<br>• There is no additional information about the side-effects |
| D. Probability information in numbers with accompanying description | Same as b, accompanied by a description of the side-effects, including whether there is something that can be done about the side-effects and whether the side-effect disappears after the treatment |
| E. Visualised probability information with accompanying description | Same as c, accompanied by a description of the side-effects, including whether there is something that can be done about the side-effects and whether the side-effect disappears after the treatment |

diagnosed patients without prior knowledge. Participants were recruited by the online panel Flycatcher (ISO-20252, ISO-27001 certified). To ensure that approximately half of the participants had low HL, women's HL was assessed before the experiments with the Set of Brief Screening Questions (SBSQ) in Dutch, a self-reported measure with three questions measured on a 5-point scale (0 = always, 4 = never) [40, 41]. Women who scored $\leq 2$ on one of the questions were indicated as having lower HL. HL questions were posed together with the question if they had (or had had) breast cancer (exclusion criteria). Randomisation with quotas on HL was used to assign women to the first or second experiment and to assign them to a presentation format. For Experiment 1, women were recruited by the Panel between August 23 and September 1, 2021, and for the second experiment between September 20 and September 30, 2021.

## Procedure

Participants received a link to an online survey through Flycatcher. Participants received hypothetical but realistic information about a woman with primary hormone-sensitive/Her2Neu-negative breast cancer and were asked to imagine themselves in the hypothetical situation. To emphasize that the medical information did not relate to the participants themselves, it was stated several times in the survey: '*Please note*: *the information is NOT real information. The information is an example. It is not about your own medical situation.*' Women then received information on survival rates (experiment 1) or survival rates and side-effects (experiment 2). Subsequently, women answered questions (see measures) while still being able to see the survival rates/side-effects information. Finally, women's numeracy and GL were assessed [42, 43]. Participants were thanked and rewarded according to the panel's agreements.

## Measures

Regarding participants' demographic characteristics, age and education level were already known to the panel. Participants' HL was measured before both experiments with the SBSQ in Dutch, a self-reported measure with three questions measured on a 5-point scale (0 = always, 4 = never). The three questions were: (1) How often do you have someone help you read

# Possible side effects of chemotherapy

### Between 1 and 10 in 100 women who receive chemotherapy experience:

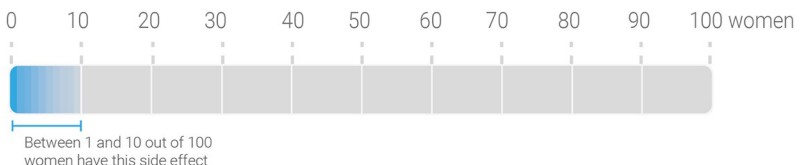

#### Sore mouth and throat, faster nosebleed
*Possible symptoms include: a dry or painful feeling in and around the mouth; sores on the gums, palate, tongue, cheeks, and lips; sensitivity to the temperature of food and drink; change or loss of taste; bleeding gums quickly; a nosebleed that stays longer.*

Is there anything that can be done about this side effect? 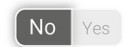

Does the side effect disappear after the treatment? 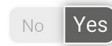

# Possible side effects of chemotherapy

### More than 10 out of 100 women who receive chemotherapy experience:

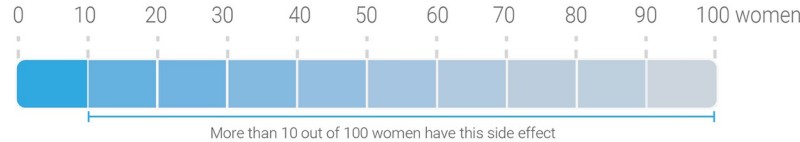

#### Prolonged fatigue
*You may feel exhausted all the time. Physical exertion may require more effort. You may also suffer from loss of concertation and memory problems. The most common complaints are: lack of energy, listlessness, less interest in the environment, and irritability*

Is there anything that can be done about this side effect? 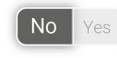
*It is recommended to keep moving.*

Does the side effect disappear after the treatment? 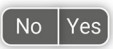
*Some of the women have to deal with chronic fatigue. For some women, the energy returns.*

**Fig 2. Example of a side-effect presentation format including visualised probability information and a description of the side-effects.**

hospital materials?; (2) How confident are you filling out medical forms by yourself?; (3) How often do you have problems learning about your medical condition because of difficulty understanding written information? [40, 41]. The SBSQ was chosen as a measure for HL because it fits our view of HL in this study, namely as an individual ability in which understanding and interpreting health information is central. Moreover, in our pre-test, people responded negatively to the NVS-D and thought it resembled a math test. The survey also included questions about the medical knowledge and medical education of participants to check afterward whether there were any differences between the groups. Table 3 provides the primary outcome measures of both experiments. The gist comprehension questions were asked before the verbatim comprehension questions. S2 File provides the secondary outcome

**Table 3. Primary outcome measures.**

| | Outcome measures | Items | Scale / response categories |
|---|---|---|---|
| **Experiment 1** | Comprehension–gist | • Which additional treatment gives the **most** benefit (extra survival)?<br>• Which choice will keep **most** women alive after 10 years?<br>• Which additional treatment gives the **least** benefit (extra survival)? | No additional treatment (option 1) / Hormone treatment (option 2) / Hormone treatment and chemotherapy (option 3) / I don't know |
| | Comprehension–verbatim | • This question is about women receiving hormone treatment (option 2). How many of the 100 women are still alive after 10 years?<br>• This question is about women receiving hormone treatment and chemotherapy (option 3). How many out of 100 women have died after 10 years?<br>• Some women receive hormone treatment (option 2). Some women do not receive additional treatment (option 1). How many more women are still alive after 10 years with option 2 compared to option 1?<br>• Some women receive hormone treatment and chemotherapy (option 3). Some women receive hormone treatment (option 2). How many more women are still alive after 10 years with option 3 compared to option 2? | . . .. women / I don't know |
| **Experiment 2** | Gist comprehension trade-off | • If you don't receive additional treatment (hormone treatment or chemotherapy), you have no chance of getting side-effects.<br>• If you receive both hormone treatment and chemotherapy, you are less likely to experience side-effects than if you receive hormone treatment alone.<br>• If you choose the treatment with the greatest chance of survival, you also have the greatest chance of side-effects. | True / False / I don't know |
| | Gist comprehension side-effects probability | • If you receive hormone treatment, you are more likely to experience *mood swings* than to experience *muscle, bone, and/ or joint pain*.<br>• If you receive chemotherapy, you are less likely to develop a *flulike feeling and muscle strain* than to develop *anaemia*.<br>• Between 1 and 10 out of 100 women receiving chemotherapy experience *deficiency of immune cells*.<br>• More than 10 out of 100 women receiving hormone treatment experience *an altered sensation in the hands and feet, such as tingling or numbness*.<br>• If you receive hormone treatment, what is **most** likely?<br>• If you receive chemotherapy, what is **least** likely? | True / False / I don't know<br>True / False / I don't know<br>True / False / I don't know<br>True / False / I don't know<br>That you suffer from *hot flushes and sweating* / *a dry vagina and less sex drive* / *a shortage of immune cells* / I don't know<br>That you get a *flulike feeling and muscle aches* / *a sore mouth and throat, nosebleeds more quickly* / *an altered sensation in your hands and feet, such as tingling or numbness* / I don't know |
| | Feeling informed | • I know which options are available to me.<br>• I know the benefits of each option.<br>• I know the risks and side-effects of each option. | 1 (strongly disagree)–<br>5 (strongly agree) |

Note. The Kuder-Richardson Reliability Coefficient for Experiment 1 is .71 for gist comprehension and .57 for verbatim comprehension. For Experiment 2 the Kuder-Richardson Reliability Coefficient is .44 for gist comprehension trade-off and .67 for gist comprehension side-effects probability. The Cronbach's Alpha of the Feeling informed scale is .84.

measures. Questions from the first experiment were pre-tested among 67 women between 18–74 years (M = 34.9; SD = 15.8) without breast cancer. In this pretest, HL was assessed using the SBSQ in Dutch [32, 33] and women scoring ≤2 on one of the questions were indicated as having lower HL skills (n = 30). To avoid ceiling effects in the experiment and to take into account respondents' comments that the number of questions made it feel like a math exam, we selected comprehension questions answered correctly by ≤90% of the women with lower HL. In addition, the number of questions was reduced because respondents indicated that the questionnaire was too long and minor textual adjustments were made if respondents indicated that the question or response category was not clear. Regarding the perception of treatment effect, questions were selected based on the two relevant comparisons (i.e., no treatment versus hormone therapy, and hormone therapy versus combined therapies).

**Primary outcome measures–Experiment 1: Presentation of survival rates.** *Comprehension–gist*. Three multiple-choice questions, related to understanding which treatment led to more/less survival. Each answer was coded as 1 (correct) or 0 (incorrect).

*Comprehension–verbatim*. Four open-ended questions, related to the exact amount of extra survival rate for the various treatments. Answers were coded as correct when they were exactly correct.

**Primary outcome measures–Experiment 2: Side-effects information in addition to survival rates.** *Gist comprehension trade-off*. Three self-composed true/false questions, with an extra 'I don't know' option. The questions were meant to address the essence of weighing the harms and benefits of the options, which was defined by the researchers as knowing that additional treatment increases chances of survival but also brings more risks of side-effects. Each answer was coded as 1 (correct) or 0 (incorrect).

*Gist comprehension probability of side-effects*. Six self-composed multiple-choice questions. Each answer was coded as 1 (correct) or 0 (incorrect). This was not calculated for format A because this format did not contain probability information.

*Feeling informed*. Three items of the Informed subscale of the Decisional Conflict Scale (DCS) on a 5-point scale (1 = strongly disagree, 5 = strongly agree) [44]. This measure was included to assess the effect of adding more detailed side-effect information on participants' feeling of being informed.

## Secondary outcome measures–Experiments 1 and 2

To measure people's feelings and subjective reactions to the presentation formats, various secondary outcomes were assessed.

**Affect.**   10 items of the Short PANAS [45]. Half of the items measure Positive Affect (PA) and half Negative Affect (NA) on a 5-point scale (1 = very slightly or not at all, 5 = extremely). Two items were added, i.e., worried and overwhelmed, based on our previous study [34].

**Hypothetical decision.**   One question about participants' choice after seeing the information.

**Decision confidence.**   One item about how confident one was about the decision (1 = not confident at all, 10 = very confident).

**Decision uncertainty [Experiment 2 only].**   Three items of the Uncertainty subscale of the DCS on a 5-point scale (1 = strongly agree, 5 = strongly disagree) [44].

**Preparedness for decision-making [Experiment 2 only].**   Six items from the Preparation for Decision-Making Scale [46]. We included the six items about decision-making and excluded four items about preparation for decision-making with a doctor in the consultation room, to match the aim of our experiment.

**Perception of treatment effect [Experiment 1 only].**   Two items measuring perception of the amount of benefit of the treatments based on Zikmund-Fisher, Angott, and Ubel [47],

measured on a 10-point scale (1 = not reduce the chance at all, 10 = reduce the chance a great deal).

**Risk perception [Experiment 2 only].** Six items about hormone therapy and six about chemotherapy, assessing perceived severity, perceived likelihood, and worry on a 10-point scale [48].

**Evaluation of the information.** Three evaluation questions with a 10-point scale (1 = totally disagree, 10 = totally agree) [49], a higher score indicates a higher evaluation.

Besides these secondary outcome measures, we used a *realism check* to verify whether participants were able to imagine themselves in the hypothetical situation, using two questions, i.e., 'The situation described was realistic' and 'I had no trouble imagining myself in this situation', with a 10-point scale (1 = totally disagree, 10 = totally agree) [48].

## Data analysis

An a priori power analysis was performed based on a 3x2 (Experiment 1) and a 5x2 factorial ANOVA design (Experiment 2) with interaction-effect using the software programs G-power and PASS. With a medium effect size (ES) of .25 (Cohen's f) [50], alpha of .05, and power of .90, the required sample sizes were 210 and 260. A medium effect size was chosen from a pragmatic point of view as we wanted to formulate recommendations for practical implementations (e.g., PtDAs). However, contrary to what was described in our pre-registration, in both experiments the analyses with comprehension as outcome were performed with cumulative odds ordinal logistic regression with proportional odds (instead of ANOVAs). This was due to the ordinal nature of the comprehension variables with limited possible outcomes (i.e., ranging from 0–3 and 0–6). Likewise, due to the categorical nature of the outcome 'hypothetical decision', the effects of format and HL on this outcome were analysed using chi-square tests of association. The other outcomes were analysed using two-way ANOVAs, with format and HL as independent variables. To account for potential effects of multiple hypothesis testing, we applied a Bonferroni correction in post hoc analyses. Results related to numeracy and GL are described in S3 File. Analyses were performed using SPSS version 26. Significance levels were set at p < .05.

## Results

### Sample characteristics

Fig 3 shows an overview of participant inclusion, non-response reasons, and randomisation. The panel examined the data for potential inattentive responders and poor data quality. Quality checks were performed on open answers, consistency of answers, straight-lining (i.e., the same answer option is chosen throughout a series of statements), and completion time, resulting in the removal of one respondent in Experiment 1 and two in Experiment 2. Randomisation with quotas on HL was used to assign women to the first or second experiment. This ensured that approximately the same number of participants with low and high HL would participate in the first and second experiments and be assigned to a presentation format.

In Experiment 1, women were on average 59.4 ± 6.1 years (N = 219), 63 (28.8%) women had a low educational level, and 98 (44.7%) had low HL. In Experiment 2, women were on average 59.6 ± 6.0 years (N = 282), 80 (28.4%) women had a low educational level, and 116 (41.1%) had low HL. Table 4 describes background characteristics of both cohorts.

### Experiment 1 –presentation of survival rates

Table 5 presents descriptive findings for the primary and secondary outcomes of Experiment 1. The ordinal logistic regression results for the interaction and main effects of format and

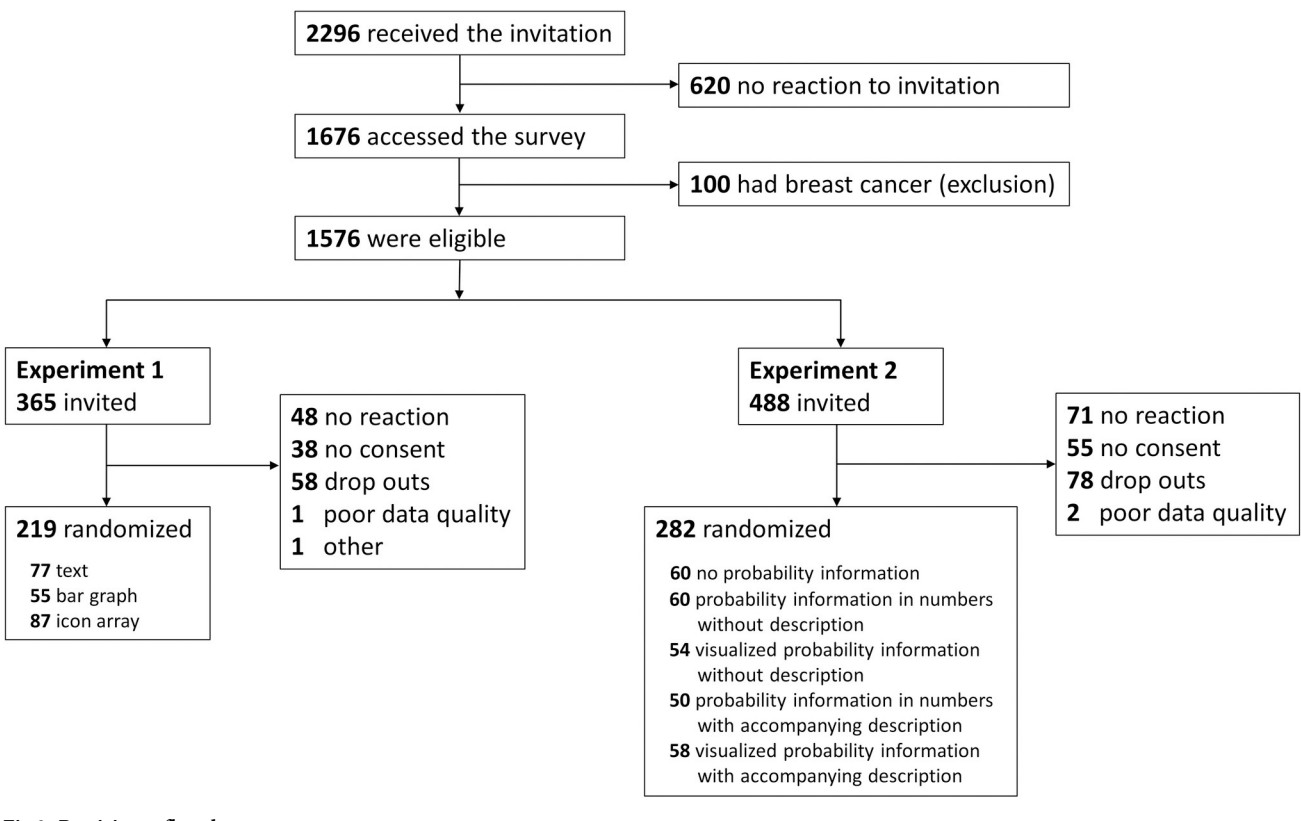

**Fig 3. Participant flowchart.**

health literacy on comprehension from both Experiments 1 and 2 described below are tabulated in S4 File.

**Primary outcomes–comprehension (H1).** Contrary to our hypothesis, there was no significant effect of presentation format (H1a), Wald $\chi^2(2) = .83$, $p = .660$, nor an interaction effect between format and HL on gist comprehension (H1c), Wald $\chi^2(2) = 2.74$, $p = .254$. HL had a significant effect on gist comprehension, Wald $\chi^2(1) = 6.84$, $p = .009$. Those with high HL exhibited a higher gist comprehension compared to those with low HL, with the odds of women with high HL having higher gist comprehension being 2.66 (95% CI, 1.28 to 5.55) times that of women with low HL.

Contrary to our hypothesis, there was no significant main effect of format on verbatim comprehension (H1b), Wald $\chi^2(2) = 2.15$, $p = .342$. Nor was HL associated with verbatim comprehension, Wald $\chi^2(1) = 1.32$, $p = .251$. The model with interaction violated the assumption of proportional odds, therefore a multinomial logistic regression was conducted to test the interaction between format and HL on verbatim comprehension. Contrary to our hypothesis (H1c), this showed no significant interaction, $\chi^2(8) = 12.35$, $p = .136$.

**Secondary outcomes.** None of the interactions or main effects exploratory tested were significant. Also, neither format, $\chi^2(4) = 6.15$, $p = .188$, nor HL, $\chi^2(2) = 3.80$, $p = .150$, were associated with the hypothetical decision. Concerning the realism check (how well women empathised with the scenario), there were no effects related to format and/or HL.

## Experiment 2 –side-effects information in addition to survival rates

Building on Experiment 1 we intended to use the best-understood survival rate format from this first experiment in Experiment 2. However, as there were no significant between-format differences, we made the pragmatic decision to continue with the text block format.

**Table 4. Sample characteristics.**

| | Format | N | Low health literacy | Age (years), mean ± SD | Education level[a] | | | Medical education[b] n (%) | Medical knowledge[c], mean ± SD (range 1–7) |
|---|---|---|---|---|---|---|---|---|---|
| | | | | | Low | Middle | High | | |
| Experiment 1 | Text block | 77 | 37 (48.1%) | 60.7 ± 6.3 | 18 (23.4%) | 42 (54.5%) | 17 (22.1%) | 14 (18.2%) | 3.1 ± 1.2 |
| | Bar graph | 55 | 24 (43.6%) | 58.8 ± 5.9 | 21 (38.2%) | 20 (36.4%) | 14 (25.5%) | 13 (23.6%) | 3.4 ± 1.3 |
| | Icon array | 87 | 37 (42.5%) | 58.6 ± 5.9 | 24 (27.6%) | 34 (39.1%) | 29 (33.3%) | 15 (17.2%) | 3.3 ± 1.3 |
| | Total | 219 | 98 (44.7%) | 59.4 ± 6.1 | 63 (28.8%) | 96 (43.8%) | 60 (27.4%) | 42 (19.2%) | 3.2 ± 1.3 |
| Experiment 2 | Format A No probability information | 60 | 26 (43.3%) | 59.6 ± 6.0 | 17 (28.3%) | 30 (50.0%) | 13 (21.7%) | 9 (15.0%) | 3.3 ± 1.2 |
| | Format B Probability information in numbers without description | 60 | 24 (40.0%) | 59.4 ± 6.5 | 19 (31.7%) | 27 (45.0%) | 14 (23.3%) | 11 (18.3%) | 3.7 ± 1.3 |
| | Format C Visualised probability information without description | 54 | 26 (48.1%) | 59.4 ± 5.8 | 15 (27.8%) | 20 (37.0%) | 19 (35.2%) | 10 (18.5%) | 3.4 ± 1.2 |
| | Format D Probability information in numbers with accompanying description | 50 | 17 (34.0%) | 58.8 ± 5.8 | 13 (26.0%) | 22 (44.0%) | 15 (30.0%) | 3 (6.0%) | 3.0 ± 1.1 |
| | Format E Visualised probability information with accompanying description | 58 | 23 (39.7%) | 60.7 ± 6.0 | 16 (27.6%) | 21 (36.2%) | 21 (36.2%) | 9 (15.5%) | 3.2 ± 1.2 |
| | Total | 282 | 116 (41.1%) | 59.6 ± 6.0 | 80 (28.4%) | 120 (42.6%) | 82 (29.1%) | 42 (14.9%) | 3.3 ± 1.2 |

Note. We checked whether there were differences in the listed characteristics between the participants of the different formats. There were no significant differences in either experiment. [a]Low education = primary education or pre-vocational secondary education; middle education = secondary vocational education, senior general secondary education, pre-university education; high education = university of applied sciences or university. [b]Medical education was assessed using one question asking whether a participant had medical, paramedical, or nursing training. [c]Medical knowledge was assessed using three questions about medical knowledge in general, knowledge about breast cancer, and knowledge about breast cancer treatments, measured on a 7-point scale (1 = no knowledge, 7 = a lot of knowledge) [51].

**Primary outcomes–comprehension (H2).** Table 6 presents descriptive findings for the primary and secondary outcomes. Contrary to the hypothesis, there was no main effect of format (H2a), Wald $\chi^2(4) = 4.68$, $p = .322$, nor a significant interaction between format and HL on gist comprehension of the trade-off (H2c), Wald $\chi^2(4) = 5.92$, $p = .206$. Nor did HL influence gist comprehension of the trade-off, Wald $\chi^2(1) = .61$, $p = .436$. Also for gist comprehension of the probability of side-effects, there were no significant effects, neither for the interaction effects of format and HL (H2c), Wald $\chi^2(3) = 1.41$, $p = .703$, nor for the main effects of format (H2b), Wald $\chi^2(3) = 1.17$, $p = .760$, or HL, Wald $\chi^2(1) = 1.56$, $p = .211$. This lack of effects was also not in line with our hypotheses.

**Primary outcomes–feeling informed (H3).** Regarding 'feeling informed' (H3), we found an interaction between HL and format, $F(4, 274) = 2.67$, $p = .032$, partial $\eta^2 = .04$. Therefore, an analysis of simple main effects was performed. For format C (visualised probability information without description), there was a difference in the average score on feeling informed between women with low and high HL, F (1,272) = 7.75, p = .006 after Bonferroni correction, partial $\eta^2 = .03$. Women with low HL presented with this format felt more informed (4.40 ±

**Table 5. Results Experiment 1 –presentation survival rates.**

| | | Text block | Bar graph | Icon array |
| --- | --- | --- | --- | --- |
| | | (n = 77) | (n = 55) | (n = 87) |
| Primary outcomes | Gist comprehension | | | |
| | 0 right answers | 1 (1.3%) | 1 (1.8%) | 4 (4.6%) |
| | 1 right answers | 2 (2.6%) | 2 (3.6%) | 7 (8.0%) |
| | 2 right answers | 9 (11.7%) | 6 (10.9%) | 5 (5.7%) |
| | 3 right answers | 65 (84.4%) | 46 (83.6%) | 71 (81.6%) |
| | Verbatim comprehension | | | |
| | 0 right answers | 6 (7.8%) | 4 (7.3%) | 2 (2.3%) |
| | 1 right answers | 3 (3.9%) | 1 (1.8%) | 5 (5.7%) |
| | 2 right answers | 2 (2.6%) | 6 (10.9%) | 11 (12.6%) |
| | 3 right answers | 34 (44.2%) | 24 (43.6%) | 43 (49.4%) |
| | 4 right answers | 32 (41.6%) | 20 (36.4%) | 26 (29.9%) |
| Secondary outcomes | Affect mean ± SD (range 1–5) | | | |
| | Positive Affect | 2.6 ± .8 | 2.5 ± .8 | 2.5 ± .7 |
| | Negative Affect | 2.2 ± 1.0 | 2.3 ± 1.1 | 2.1 ± 1.0 |
| | Perception of treatment effect mean ± SD (range 1–10) | | | |
| | Hormone therapy | 6.6 ± 1.8 | 6.5 ± 1.9 | 6.5 ± 1.5 |
| | Hormone therapy and chemotherapy | 6.8 ± 2.1 | 6.7 ± 1.9 | 6.7 ± 1.6 |
| | Evaluation of information mean ± SD (range 1–10) | 7.7 ± 1.9 | 7.3 ± 1.6 | 7.4 ± 1.8 |
| | Hypothetical decision, n (%) | 15 (19.5%) | 7 (12.7%) | 8 (9.2%) |
| | No additional treatment | 14 (18.2%) | 17 (30.9%) | 20 (23.0%) |
| | Hormone therapy | 48 (62.3%) | 31 (56.4%) | 59 (67.8%) |
| | Hormone therapy and chemotherapy | | | |
| | Decision confidence mean ± SD (range 1–10) | 6.9 ± 2.3 | 6.8 ± 2.1 | 6.8 ± 2.0 |

.64) than women with high HL presented with this format (3.89 ± .73), a mean difference of .51 (95% CI, .15 to .86). The interaction is displayed in Fig 4. Other simple main effects were not significant.

**Secondary outcomes.** There was a main effect of HL on Negative Affect (PANAS NA), $F(1, 272) = 5.80$, $p = .017$, partial $\eta^2 = .02$. Women with low HL experienced more Negative Affect (marginal means 2.60 ± .11) than women with high HL (marginal means 2.27 ± .09), a mean difference of .34 (95% CI, .06 to .61). Interactions or main effects for the other affect outcomes were not significant.

For risk perception regarding hormone treatment, there was a main effect for format, $F(4, 272) = 4.40$, $p = .002$, partial $\eta^2 = .06$. The pairwise comparisons showed a significant difference between format A (no probability information) and format C (visualised probability information without description) of .91 (95% CI, .24 to 1.58), p = .002. Risk perception for women presented with format A (no probability information) was higher (marginal means 7.34 ± .16) compared to risk perception of women presented with format C (visualised probability information without description; marginal means 6.44 ± .17). For risk perception regarding chemotherapy, the ANOVA showed a main effect for format, $F(4, 272) = 2.41$, $p = .049$, partial $\eta^2 = .03$, but none of the post hoc pairwise comparisons were statistically significant.

For the secondary outcomes decision uncertainty, preparedness for decision-making, evaluation of information, and decision confidence, there were no interactions or main effects. The effects of format and HL on the hypothetical decision were also not significant, nor were interactions or main effects regarding the realism check.

**Table 6. Results Experiment 2 –side-effects information in addition to survival rates.**

| | | Format A | Format B | Format C | Format D | Format E |
|---|---|---|---|---|---|---|
| | | **No probability information** | **Probability information in numbers without description** | **Visualised probability information without description** | **Probability information in numbers with accompanying description** | **Visualised probability information with accompanying description** |
| | | **(n = 60)** | **(n = 60)** | **(n = 54)** | **(n = 50)** | **(n = 58)** |
| **Primary outcomes** | Gist comprehension trade-off | | | | | |
| | 0 right | 0 (0%) | 3 (5.0%) | 5 (9.3%) | 5 (10%) | 3 (5.2%) |
| | 1 right | 12 (20.0%) | 10 (16.7%) | 12 (22.2%) | 10 (20%) | 11 (19.0%) |
| | 2 right | 17 (28.3%) | 20 (33.3%) | 18 (33.3%) | 14 (28.0%) | 21 (36.2%) |
| | 3 right | 31 (51.7%) | 27 (45.0%) | 19 (35.2%) | 21 (42.0%) | 23 (39.7%) |
| | Gist comprehension probability of side-effects | | | | | |
| | 0 right | | 3 (5.0%) | 6 (11.1%) | 1 (2.0%) | 4 (6.9%) |
| | 1 right | | 9 (15.0%) | 8 (14.8%) | 8 (16.0%) | 6 (10.3%) |
| | 2 right | | 7 (11.7%) | 4 (7.4%) | 12 (24.0%) | 6 (10.3%) |
| | 3 right | | 19 (31.7%) | 11 (20.4%) | 5 (10.0%) | 15 (25.9%) |
| | 4 right | | 6 (10.0%) | 8 (14.8%) | 11 (22.0%) | 7 (12.1%) |
| | 5 right | | 13 (21.7%) | 9 (16.7%) | 9 (18.0%) | 12 (20.7%) |
| | 6 right | | 3 (5.0%) | 8 (14.8%) | 4 (8.0%) | 8 (13.8%) |
| | | | | | | |
| | Feeling informed | 4.1 ± .6 | 4.0 ± .7 | 4.1 ± .7 | 4.0 ± .7 | 4.1 ± .6 |
| | mean ± SD (range 1–5) | | | | | |
| **Secondary outcomes** | Affect mean ± SD (range 1–5) | | | | | |
| | Positive Affect | 2.4 ± .6 | 2.4 ± .7 | 2.4 ± .6 | 2.3 ± .7 | 2.4 ± .7 |
| | Negative Affect | 2.4 ± 1.2 | 2.3 ± 1.1 | 2.5 ± 1.1 | 2.3 ± 1.2 | 2.6 ± 1.2 |
| | Decision uncertainty | 3.4 ± .8 | 3.2 ± .8 | 3.2 ± 1.1 | 3.4 ± .9 | 3.1 ± .8 |
| | mean ± SD (range 1–5) | | | | | |
| | Preparedness for decision-making | 3.8 ± .8 | 3.7 ± .8 | 3.8 ± .7 | 3.8 ± .7 | 3.8 ± .6 |
| | mean ± SD (range 1–5) | | | | | |
| | Risk perception | | | | | |
| | mean ± SD (range 1–10) | | | | | |
| | Hormone therapy | 7.3 ± 1.0 | 7.0 ± 1.2 | 6.4 ± 1.3 | 6.6 ± 1.3 | 6.8 ± 1.4 |
| | Chemotherapy | 8.4 ± .9 | 8.0 ± 1.3 | 7.9 ± 1.3 | 7.8 ± 1.4 | 8.3 ± 1.1 |
| | Evaluation of information | 7.8 ± 1.5 | 7.7 ± 1.5 | 7.5 ± 1.4 | 7.8 ± 1.6 | 7.6 ± 1.7 |
| | mean ± SD (range 1–10) | | | | | |
| | Hypothetical decision, *n* (%) | | | | | |
| | No additional treatment | 18 (30.0%) | 14 (23.3%) | 12 (22.2%) | 9 (18.0%) | 9 (15.5%) |
| | Hormone therapy | 13 (21.7%) | 15 (25.0%) | 17 (31.5%) | 17 (34.0%) | 15 (25.9%) |
| | Hormone therapy and | 29 (48.3%) | 31 (51.7%) | 25 (46.3%) | 24 (48.0%) | 34 (58.6%) |
| | Chemotherapy | | | | | |
| | Decision confidence | 6.8 ± 2.0 | 6.3 ± 2.1 | 6.4 ± 2.3 | 6.7 ± 2.0 | 6.1 ± 2.2 |
| | mean ± SD (range 1–10) | | | | | |

Note. Regarding format A 'No probability info and no accompanying description' the gist comprehension probability of side-effects was not calculated because this information was not presented in this format.

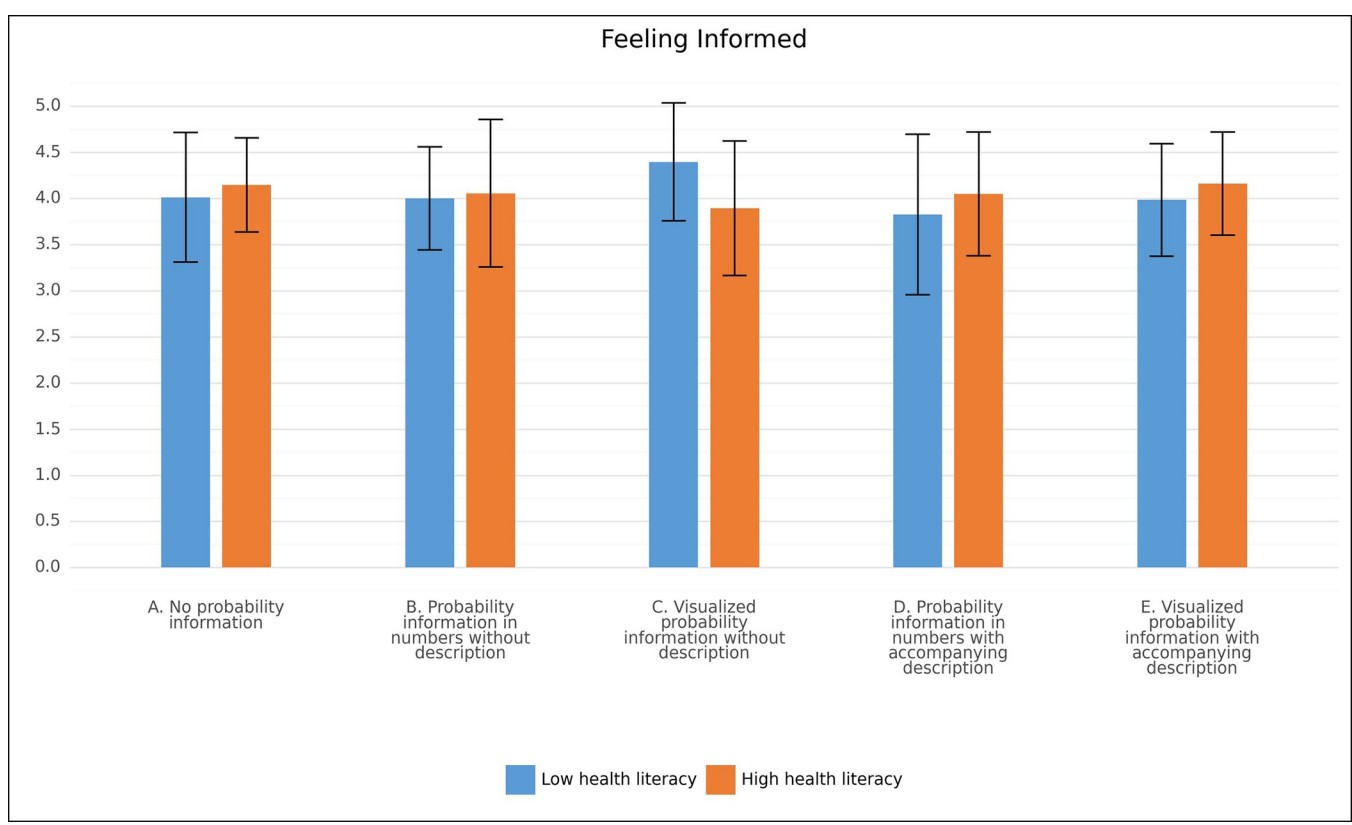

**Fig 4. Interaction effect of side-effect format and health literacy.** Note. For format C (visualised probability information without description), there was a difference in the average score on feeling informed between women with low and high HL. As can be seen, although this was not a significant difference, for women with low HL, this specific format evoked the highest feelings of being informed, whereas for women with high HL, this specific format evoked the lowest feelings of being informed.

## Discussion

In this study, we investigated the effects of several (visual) presentation formats to present decision-relevant numerical information (i.e., survival rates and side-effects) to patients in support of informed and shared decision making (SDM). Results showed that, based on medium effect sizes, different well-designed presentation formats that adhere to best practices in probability communication did not differ in terms of comprehension of the end-users. However, regardless of presentation format, women with low health literacy (HL) exhibited worse gist understanding of the survival rates than women with high HL. Regarding the side-effects formats, when the format with visualised probability information without a description of the specific side-effects was shown, women with low HL felt better informed than women with high HL.

There may be several explanations for the lack of beneficial effects of the visual formats compared to a text format on respondents' understanding. First, it might be that the text blocks showing the numerical information were quite optimal because they followed general best practices in probability communication as well as graphical design principles. Previous studies found that structuring textual information in, for example, fact boxes or tables can lead to the same or better level of comprehension compared to non-structured textual formats or other graphical displays [25, 52–54]. The advantage of structured textual formats can be that, unlike bar graphs and icon arrays, no legend needs to be interpreted. It is also possible that

despite developing the visualisations in co-creation with the target group, the visualisations were still not optimal. For example, women who were particularly focused on the visualisation and the legend may not have noticed that all the options were about someone who had already undergone surgery. It can also be argued that the relatively limited additional benefit of hormone therapy and chemotherapy (4% and 5%, respectively) versus a relatively large percentage of people surviving without treatment (72%) may have played a role in the lack of effects of the visual formats. It might be that other survival-to-benefit ratios may be more noticeable when displayed visually.

Another potential explanation may be related to the difficulty of the information. For example, when developing the survival rate formats with patients, women expressed a need for an overview of options, including 'no additional treatment'. This resulted in formats with three options, which may have been overwhelming, especially for those with lower information processing skills. Indeed, our study showed that women with low HL had worse understanding of the gist of survival rates than women with high HL. A study on the same three treatment options showed that comprehension increased when presenting options as two decisions instead of one (no additional treatment versus receiving hormone therapy and then hormone therapy versus hormone therapy with chemotherapy) [47]. It may be worthwhile to further explore how women's expressed need for an overview of treatment options can be combined with sequential presentation of information.

Also regarding the side-effects, the difficulty of the information may have played a role. The information about side-effects was generally not well understood, both by people with higher and lower HL. Dividing the probabilities into two categories (i.e., occurrence in more than 10 out of 100 women and occurrence in 1 to 10 out of 100 women) might be more difficult to interpret and compare than exact probability information (e.g., 8 out of 100 women will experience this side-effect). Additionally, more than 10 out of 100 women represents a wide range of probabilities. This probably makes the information, even when visualised, more difficult. Exact point estimates were not available, which raises the question of how to deal with this in practice. A limited amount of research has investigated the presentation of uncertainty in icon arrays with colour gradient, shading, or arrow, but either the effect on comprehension was not (yet) examined, or no differences were found compared to no visualisation [55, 56]. Further research into how to communicate a range is therefore needed.

Regardless of format, women with lower HL experienced more negative affect than women with high HL. However, when provided with visualised probability information without a description of the side-effects, women with low HL felt better informed than women with high HL. An explanation might be that those with lower HL might not accurately estimate how informed they are, as found in previous research [57]. However, we found significant correlations between the scores for knowledge and feeling informed for both women with lower and higher HL. This might be because we, unlike the previous study, measured these outcomes immediately after information provision. Also concerning risk perception related to hormone treatment, an effect for visualising probability information was found. Women provided with visualised probability information without a description of side-effects exhibited a lower risk perception than those provided with no probability information. However, this effect was not present in the format containing a description of the side-effects. An explanation could be that the amount of information in this description reduced the positive effect of the visualisation. In these cases, it may be that less information (e.g., not a description for all the side-effects) may be preferred by lower HL people to gain a sense of 'mastery' of this complex information.

When examining comprehension, gist and verbatim comprehension were assessed using self-composed questions. This was due to an absence of validated questionnaires, as these concepts depend on the specific information presented. Especially for gist comprehension, it

remains difficult to assess which gist representations count as 'accurate' and how surveys should capture this [58]. We reasoned that the essential information was which additional treatment gives the most benefit (extra survival, Experiment 1) and that additional treatment increases survival but also brings more risks of side-effects (Experiment 2). One may argue that other, unassessed, gist representations can also be distilled from the information, such as that additional benefit of adjuvant systemic therapy is 'relatively small' or that even without additional treatment most women will still be alive in 10 years. However, women with low HL exhibited worse gist understanding overall, suggesting that we captured at least some important aspects of gist representations. It should be noted that the comprehension questions were mainly aimed at understanding the core message of the information. This may have resulted in the comprehension questions in Experiment 1 in particular being too easy and not necessarily the most focused on discovering differences between the formats. Therefore, the comprehension questions themselves may also have contributed to the lack of effects. The survey as a whole may also have played a role, as women had to understand not only the information but also the questions. The questionnaire was translated to a reading level of up to sixth grade by a plain language expert and participants saw the information on screen while answering questions. Nevertheless, the questions may have been difficult, especially for those with lower HL.

This study used adjuvant systemic therapy for breast cancer as a case example. However, more and more decision-relevant data are becoming available for other forms of cancer as well, such as lung cancer and stomach and oesophageal cancer. The results of the current study can be important in presenting decision-relevant numerical information more broadly in oncology. However, the context and specific decision to be made should be taken into account. For example, the survival rates, available treatment options, prognosis, and the average age of the patient population can influence comprehension. User-testing within the specific context remains necessary.

## Strengths and limitations

A strength of this study is the inclusion of information about both survival rates and side-effects of treatment options. Previous studies examined visual formats of survival rates for adjuvant systemic breast cancer treatment [26, 27], but without the comparison with a textual format. Other studies investigated side-effects message/presentation format [e.g., 29, 52, 59], but not in combination with survival rate information. For future research, it may be interesting to investigate whether the different combinations of survival rates and side-effect formats have different effects on the trade-off to be made, as the different combinations of formats were not investigated in the current study.

A potential limitation regarding generalizability in practice is the use of hypothetical scenarios. It may be that respondents paid less attention to the information due to the hypothetical scenario. Besides, the information is meant for women diagnosed with breast cancer, a stressful and life-threatening situation involving emotions, which may influence information processing. Another potential limitation is that we do not know what previous experiences our respondents had with cancer. Moreover, multiple (secondary) outcome measures were examined, resulting in multiple testing. However, since no major effects were found, this limitation ultimately did not influence our findings.

The effect sizes of the studies on which the International Patient Decision Aid Standards (IPDAS) collaboration's recommendation to use visualisations is based are generally small to moderate [4]. Sample sizes in our experiments were based on medium effect sizes of Cohen's f .25, a pragmatic choice as our starting point was to make recommendations for implementation in practice (e.g., PtDAs). However, these medium effect sizes may be a reason for the lack

of differences between formats and it should be noted that non-significant findings do not necessarily indicate the true absence of an effect. Moreover, initial power calculations were based on factorial ANOVA designs, whereas ultimately ordinal logistic regression analyses were performed. This may have affected the power. In addition, based on the initial power calculations, group sizes for the bar graph format in Experiment 1 were smaller for the low HL (n = 24) and high HL (n = 31) than the required n = 35 for a 90% power to detect an interaction-effect based on a medium effect size. For Experiment 2, group sizes were smaller for the low HL presented with format B (n = 24), format D (n = 17), and format E (n = 23) than the required n = 26 for a 91% power to detect an interaction-effect based on a medium effect size. However, it should be noted that the expected interaction-effects were ordinal-interactions rather than full crossover interactions, therefore the statistical power to detect the expected interactions is lower than the a priori calculated 90% and 91%. This implies that the question of whether people with lower health literacy levels would benefit more from the formats with the visualizations than people with higher health literacy could not be answered reliably.

## Conclusion

No evidence was found for a medium effect size in comprehension when presenting decision-relevant numerical information to patients in either a well-designed text block, bar graph, or icon array that all adhered to risk communication best practices. Providing patients with visualisations might not necessarily yield an advantage over providing structured numerical information. These results have practical implications, for example, for patient decision aid developers. The fact that visualising numerical information is not a magic bullet is relevant when developing patient decision aids. Furthermore, the results of this study show that a deeper understanding of how to present numerical and context-specific information about side-effects seems needed. Especially for patients with lower information processing skills this is important. They understood the information less well and experienced more negative affect when receiving side-effect information.

## Supporting information

**S1 File. Side-effects formats.**
(PDF)

**S2 File. Table secondary outcomes.**
(PDF)

**S3 File. Numeracy and graph literacy.**
(PDF)

**S4 File. Ordinal logistic regression for the interaction and main effects of format and health literacy on comprehension.**
(PDF)

## Acknowledgments

We would like to thank Maaike Weber for designing the presentation formats, Dr. Peter van de Ven for his support in the a priori power analysis and Dr. Birgit Lissenberg-Witte for her advice on the analyses. Also, all women who participated in the pretest or the experiments are thanked for their participation. Finally, we would like to thank PATIENT+, PacMed, the Netherlands Comprehensive Cancer Organization (IKNL), and the Dutch Breast Cancer Association (BVN) for their collaboration in the consortium entitled 'Personalised decision support

systems in breast cancer care: integrating prediction modelling with user-centred research' of which this study was part.

## Author Contributions

**Conceptualization:** Inge S. van Strien-Knippenberg, Daniëlle R. M. Timmermans, Ellen G. Engelhardt, Inge R. H. M Konings, Olga C. Damman.

**Formal analysis:** Inge S. van Strien-Knippenberg.

**Funding acquisition:** Olga C. Damman.

**Investigation:** Inge S. van Strien-Knippenberg.

**Methodology:** Inge S. van Strien-Knippenberg, Daniëlle R. M. Timmermans, Ellen G. Engel-hardt, Inge R. H. M Konings, Olga C. Damman.

**Supervision:** Daniëlle R. M. Timmermans, Olga C. Damman.

**Writing – original draft:** Inge S. van Strien-Knippenberg.

**Writing – review & editing:** Daniëlle R. M. Timmermans, Ellen G. Engelhardt, Inge R. H. M Konings, Olga C. Damman.

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
