## [Decision Letter · Decision Letter 0]

2 Jan 2024

PONE-D-23-32983Presenting decision-relevant numerical information concerning harms and benefits to patients with varying levels of Health Literacy: case example of adjuvant systemic therapy for breast cancerPLOS ONE

Dear Dr. van Strien,

Thank you for submitting your manuscript to PLOS ONE. After careful consideration, we feel that it has merit but does not fully meet PLOS ONE’s publication criteria as it currently stands. Therefore, we invite you to submit a revised version of the manuscript that addresses the points raised during the review process. Please submit your revised manuscript by Feb 16 2024 11:59PM. If you will need more time than this to complete your revisions, please reply to this message or contact the journal office at plosone@plos.org. Please include the following items when submitting your revised manuscript:A rebuttal letter that responds to each point raised by the academic editor and reviewer(s). You should upload this letter as a separate file labeled 'Response to Reviewers'.A marked-up copy of your manuscript that highlights changes made to the original version. You should upload this as a separate file labeled 'Revised Manuscript with Track Changes'.An unmarked version of your revised paper without tracked changes. You should upload this as a separate file labeled 'Manuscript'.

We look forward to receiving your revised manuscript.

Kind regards,

Felix G. Rebitschek

Academic Editor

PLOS ONE

3. In this instance it seems there may be acceptable restrictions in place that prevent the public sharing of your minimal data. However, in line with our goal of ensuring long-term data availability to all interested researchers, PLOS’ Data Policy states that authors cannot be the sole named individuals responsible for ensuring data access (http://journals.plos.org/plosone/s/data-availability#loc-acceptable-data-sharing-methods).

Reviewers' comments:

Reviewer's Responses to Questions

**Comments to the Author**

1. Is the manuscript technically sound, and do the data support the conclusions?

Reviewer #1: Yes

Reviewer #2: Partly

Reviewer #3: Yes

2. Has the statistical analysis been performed appropriately and rigorously? 

Reviewer #1: Yes

Reviewer #2: Yes

Reviewer #3: Yes

3. Have the authors made all data underlying the findings in their manuscript fully available?

Reviewer #1: Yes

Reviewer #2: No

Reviewer #3: No

4. Is the manuscript presented in an intelligible fashion and written in standard English?

Reviewer #1: Yes

Reviewer #2: Yes

Reviewer #3: Yes

5. Review Comments to the Author

Reviewer #1: This paper reports a study which aimed to investigate the effect of survival rates and side-effects presentation formats on understanding and feeling informed and effects of health literacy using breast cancer therapy as an example. The study question is important and the study is interestingly designed. I do however have a few important comments for the authors to consider:

- Overall, the writing could be clearer. For example, some of the sentences are quite long.

- Why did the authors choose breast cancer therapy as the case example? Could findings be different in another example? This is not entirely clear to me throughout.

- There is a lot of information in the Methods – it is quite hard to follow. I appreciate all of the Tables and Figures, however is there any way this could be simplified? Consider collating all of it together as an Appendix (ie. the entire survey) and simplifying in the text to make it easier for those who are not familiar with these methods to follow.

- Following on from that, there are a lot of tables – I think some need to be reduced (suggest in the Methods as opposed to the Results).

- In the first Figure make clear Experiment 1 and 2.

- The figures are coming up blurry – please check quality.

- While the Discussion is well written, the conclusions could be much stronger. What does this mean for current policy and practice – what else could be done to help with this or future research in regard to health literacy?

Reviewer #2: This study endeavors to create and evaluate visual aids conveying both survival rates and the probabilities of side effects associated with a specific breast cancer treatment. The design of visual aids adhered to state-of-the-art recommendations and involved consultations with the oncologist and patients. The manuscript maintains a robust theoretical foundation, exhibiting transparency through preregistration, comprehensive outcome reporting, and inclusion of statistical analyses, even when non-significant.

However, to enhance the manuscript's accessibility for other researchers, I propose several improvements. Firstly, supplementing details in the supplementary materials compensating for the lack of a dataset would increase the potential utility of the research in future meta-analyses (e.g., by incorporating a matrix of correlations and reliability analyses for all measures in the supplementary materials).

Additionally, a more comprehensive dropout analysis for each experimental condition, along with updates to Figure 1, is recommended, considering potential variations in emotional responses across conditions. Figure 1 would also benefit from additional information on the assignment method for experiments.

Furthermore, I suggest conducting analyses controlling for numeracy, graph literacy, and education to ensure that health literacy indeed predicts the outcomes. As the study could evoke negative emotions, please provide more details about measures that the authors used to ensure the well-being of the subjects.

I recommend, adding more information regarding the numeracy measure, including the reference, psychometric properties, and relationships with other measures (the measure is introduced only briefly in the supplementary materials).

Finally, A revised title reflecting the exclusive focus on female subjects aged 50-70 and the absence of real breast cancer patients would enhance accuracy.

Reviewer #3: This manuscript reports an online experimental study conducted among women in the Netherlands. The study investigates important questions that are broadly relevant for determining the best formats to use in health information provision. The manuscript is generally clear and well written. Below, I have indicated a few points where I think information is missing and some suggestions to consider to improve the manuscript’s presentation and English expression.

DATA AVAILABILITY

1. Information from the manuscript submission system says the “study was exempted from review by the medical research ethics committee”. However, it also says: “Data cannot be shared publicly because the Ethics Committee approved the collection and analysis of the data for the specific study only. The Ethics Committee requires that the data collected remains securely stored and not be shared publicly.” To me, these statements seem contradictory. The authors need to clarify.

INTRODUCTION

2. Page 5 line 97-98. This sentence is too long and awkward. I suggest breaking it down into 2 sentences to convey the ideas that (a) unlike previous studies, we designed info to reflect practice; (b) specifically, in practice info is complex and includes… etc.

METHODS

3. Page 8 line 145. I would save the presentation and mention of Figure 1 until the Results section.

4. Page 8 line 149-156. I think the information about the sample size calculation belongs in the Data Analysis section.

5. Please specify when and how participant demographics and background characteristics were collected (i.e., age, education level, medical background, medical knowledge).

6. Page 9 line 163. “empathise with this situation” may be the accurate direct translation from Dutch, but reading the scenario presented in Figure 2, I suggest it would be more appropriate to say “imagine themselves in the hypothetical situation” (as per Page 15 line 259).

7. Table 4. The abbreviations PA and NA are defined in the text, but they should also be in a footnote to the table. Also, this label is missing from Overwhelmed and Worried. After reading further, I now realise this is because these 2 terms were added to the original PANAS items. However, I think it would be helpful to use an asterisk or something and put a brief explanation in a footnote to the table.

8. Secondary outcome measures. To avoid confusion with “decision uncertainty”, I suggest renaming the single item “decision confidence” instead of “decision certainty”.

RESULTS

9. Table 5. A cell in the middle of this table – “20 (37.0)” – is missing the “%” sign.

10. Did the authors check for any significant differences between groups on any of these characteristics? If there were differences, should these have been controlled for in analyses?

11. Page 18 line 297. “HL was neither associated” should be “Nor was HL associated”.

12. Page 19 line 316. Similarly, “HL neither influences” should be “Nor did HL influence”.

13. Figure 4. I think the authors should consider whether this information would be better conveyed via a bar chart, because there are distinct groups/conditions being compared (as opposed to different points in time).

DISCUSSION

14. Page 26 line 449. “this limitation does not seem to have occurred” doesn’t seem quite right; perhaps replace with something like “this limitation was not consequential” or “this limitation ultimately did not influence our findings”.

15. Page 26 line 450. The acronym IPDAS should be explained.

16. Page 26 line 456. “while ordinal logistic regression analyses were performed” – I suggest instead “whereas ultimately ordinal logistic regression analyses were performed”.

6. PLOS authors have the option to publish the peer review history of their article (what does this mean?). If published, this will include your full peer review and any attached files.

Reviewer #1: No

Reviewer #2: No

Reviewer #3: **Yes: **Dr Jolyn Hersch

---

## [Author Response · Author response to Decision Letter 0]

23 Feb 2024

Dear editors,

Thank you for the opportunity to resubmit a revised version of our manuscript (Submission ID PONE-D-23-32983). We appreciate the constructive feedback and useful suggestions of the reviewers. We have addressed all reviewers’ concerns as described in the Response to reviewers File. Each review comment is described and followed by our response. In our answers, changes to the manuscript are underlined. The main changes in the revised manuscript itself are indicated by tracked changes. The line numbers in our responses to the reviewers refer to the line numbers in the tracked changes manuscript.

Regards, 

Inge van Strien

---

## [Decision Letter · Decision Letter 1]

10 Apr 2024

PONE-D-23-32983R1Presenting decision-relevant numerical information to Dutch women aged 50-70 with varying levels of Health Literacy: case example of adjuvant systemic therapy for breast cancerPLOS ONE

Dear Dr. van Strien,

Thank you for submitting your manuscript to PLOS ONE. After careful consideration, we feel that it has merit but does not fully meet PLOS ONE’s publication criteria as it currently stands. Therefore, we invite you to submit a revised version of the manuscript that addresses the points raised during the review process.

We look forward to receiving your revised manuscript.

Kind regards,

Felix G. Rebitschek

Academic Editor

PLOS ONE

Additional Editor Comments :

Dear Authors,

Thank you for submitting a revision that has addressed the reviewers’ concerns well!

My outstanding points are listed below:

1) The presentation formats fit for groups of differential education, numeracy, or health literacy is an important target. However, an argumentation is required. Why do you expect – according to which theory of health literacy – that people with different levels of health literacy would respond differently to a given piece of health information? For instance, HL-EU substantially refers to the perceived ability to seek, to find, and apply health information (besides evaluating and comprehending). For instance, the sentence in l.39-40, according to which skills related to information processing could be captured by HL is too speculative. The introduction needs to derive the health-literacy-expectations from the literature. Why should which concept of HL produce which type of differences given formats? This also may help explain why HL made no difference in comprehension in Experiment 2.

2) Your descriptive statistics even with a higher powered sample let nobody expect to find medium effects on comprehension depending on health literacy. Provided an argumentation for an interaction with health literacy, this leads to discuss

a. the stimuli: The icon array and the bar chart in Experiment 1 have a small but relevant error how visualisations are labelled: The legend labels Hormone therapy and Chemotherapy in option 2 and 3 although correctly would be Surgery and Hormone Therapy and Surgery and Chemotherapy, respectively. A scientist may perceive this difference trivial, but actually this misspecification can create confusion in laypeople who aim to understand the legend, particularly if they are not really aware about that they might have already undergone surgery. Items referring to Hormone and Chemotherapy (verbatim!) could bring the visualisation formats at disadvantage. On the other hand, it would be sufficient to read (in Experiment 1) the statement “Alive after 10 years: XX%” – this allows for correct responding to any item and this is constant across presentation conditions. So, why to expect any difference? One could say, the graphs did not mislead them.

b. the participants: Have they just not taken deep notice of the material and answered the questions correctly anyway? [e.g., the gist comprehension questions] Where attention checks for compliant responding included [only three for “poor data quality”, e.g. straightliners, excluded?]? I comprehend the power calculation, but the cells with low literates in the end are not all sufficiently powered, even assuming highly compliant participants.

c. the measurement. You did pretesting with 30 low-literate participants, but what has been learned? Items with more than 90% correctness in their group did not enter the main survey, anything else on discriminability? Probably the three gist comprehension tasks were too easy for guessing people – many people would expect more treatment more benefit. What are the internal consistencies of comprehension gist and comprehension verbatim and comprehension combined?

d. the analysis. Have you considered format analysis across all items (simple comprehension sum score)? Have you considered a sensitivity analysis excluding those, who respond “I don’t know”?

3) Please refer not only to shared decision making but to the goal of health communication enabling informed decisions according to evidence-based medicine, Western health system standard. Particularly, informing patients about benefits and harms is one of many rules according to established guidelines on how to design health communication (e.g., .

4) Also, how did you arrive at subscale of decisional conflict (instead of the full assessment?) Could you derive in the introduction why it is relevant to assess how someone could have felt informed?

5) Abstract

a. “When communicated adequately..”

b. Capitalisation of shared decision making and health literacy seems unusual

c. When high/low …. Perhaps better expressed “depending on their”….

d. Probability information in numbers/visualisations … Perhaps better expressed “numbers with or without…” Numbers accompanied also visualisations here.

6) Introduction; generally: the impression of specific visualisation that outperforms no visualisation should be avoided, because state of evidence is that different presentation formats are beneficial for different problems and different dialog groups. Please leave it in a format comparison, as you analysed it, bar vs. text and icon vs. text.

a. L.49… reduced by conveying the part-to-whole relationship [11].

b. L.62… please be more explicit what is meant by general recommendations. There is evidence for different formats and decision problems, but do you mean guidelines?

c. L.61-l.76: some studies could be considered that compared communication formats for medical evidence (with and without text control) with regard to knowledge/comprehension, also with regard to education and health literacy – references below; you may find further literature, if you review literature on the health communication of benefits and harms

d. What is meant by decision (l.95) for women without BC from the general population (hypothetical decision, intention?).

e. Experiment 1 and 2 should be capitalised throughout the manuscript

7) Method

a. Please mention that it is a convenience sample, not representative for Dutch women from 50-70.

b. Move the SBSQ to the Measures section

c. Why SBSQ, please explain, given so many others?

d. Though mentioned in the text, I cannot recognise quotas from Fig. 3.

e. How have you excluded that participants of Experiment1 participate in the subsequent Experiment2?

f. Which questions about medical knowledge and medical education have been included (reference), where reported, and why at all?

g. 1 decimal might be sufficient for age M and SD

h. What was the order of the comprehension items?

8) Results

a. Figures that illustrate the main effects and (non-)interactions with regard to gist and verbatim comprehensions would be very helpful.

b. Figure 4

i. Commas on the y-axis!?

ii. The figure showing sample-based data requires uncertainty intervals.

c. What is low, middle, high education?

d. Table 5 does not need a total column

e. Two decimals for test values and confidence intervals might be sufficient (APA)

f. L.463 (you pointed above on Bonferroni adjustments with regard to the large number of secondary outcome analyses), but here risk perception is considered to reveal an effect – please check across the results whether you corrected as planned.

9) Discussion

a. Why women with low HL did comprehend less about survival rates but not less about side effects?

10) References

a. See 13 and 45, there are variations across the references in capitalisation and abbreviations of journals, please ensure references consistency!

References to the Editor’s comment

Brick, C., McDowell, M., & Freeman, A. L. (2020). Risk communication in tables versus text: a registered report randomized trial on ‘fact boxes'. Royal Society Open Science, 7(3), 190876.

Hinneburg, J., Lühnen, J., Steckelberg, A., & Berger-Höger, B. (2020). A blended learning training programme for health information providers to enhance implementation of the Guideline Evidence-based Health Information: development and qualitative pilot study. BMC Medical Education, 20, 1-11.

McDowell, M., Gigerenzer, G., Wegwarth, O., & Rebitschek, F. G. (2019). Effect of tabular and icon fact box formats on comprehension of benefits and harms of prostate cancer screening: a randomized trial. Medical Decision Making, 39(1), 41-56.

Scalia, P., Schubbe, D. C., Lu, E. S., Durand, M. A., Frascara, J., Noel, G., ... & Elwyn, G. (2021). Comparing the impact of an icon array versus a bar graph on preference and understanding of risk information: Results from an online, randomized study. Plos one, 16(7), e0253644.

Reviewers' comments:

Reviewer's Responses to Questions

**Comments to the Author**

1. If the authors have adequately addressed your comments raised in a previous round of review and you feel that this manuscript is now acceptable for publication, you may indicate that here to bypass the “Comments to the Author” section, enter your conflict of interest statement in the “Confidential to Editor” section, and submit your "Accept" recommendation.

Reviewer #1: All comments have been addressed

Reviewer #3: All comments have been addressed

2. Is the manuscript technically sound, and do the data support the conclusions?

Reviewer #1: Yes

Reviewer #3: Yes

3. Has the statistical analysis been performed appropriately and rigorously? 

Reviewer #1: I Don't Know

Reviewer #3: Yes

4. Have the authors made all data underlying the findings in their manuscript fully available?

Reviewer #1: Yes

Reviewer #3: (No Response)

5. Is the manuscript presented in an intelligible fashion and written in standard English?

Reviewer #1: Yes

Reviewer #3: Yes

6. Review Comments to the Author

Reviewer #1: The authors have done a good job at addressing all of the previous comments and concerns and have made some extensive changes. The manuscript is much clearer. I have no further comments.

Reviewer #3: (No Response)

7. PLOS authors have the option to publish the peer review history of their article (what does this mean?). If published, this will include your full peer review and any attached files.

Reviewer #1: No

Reviewer #3: **Yes: **Dr Jolyn Hersch

---

## [Author Response · Author response to Decision Letter 1]

27 May 2024

Additional Editor Comments:

Dear Authors,

Thank you for submitting a revision that has addressed the reviewers’ concerns well!

My outstanding points are listed below:

1) The presentation formats fit for groups of differential education, numeracy, or health literacy is an important target. However, an argumentation is required. Why do you expect – according to which theory of health literacy – that people with different levels of health literacy would respond differently to a given piece of health information? For instance, HL-EU substantially refers to the perceived ability to seek, to find, and apply health information (besides evaluating and comprehending). For instance, the sentence in l.39-40, according to which skills related to information processing could be captured by HL is too speculative. The introduction needs to derive the health-literacy-expectations from the literature. Why should which concept of HL produce which type of differences given formats? This also may help explain why HL made no difference in comprehension in Experiment 2.

RESPONSE: We thank the editor for this excellent comment. We agree that more elaboration of the HL concept used in our study is useful. In our line of thinking/expectations, those who have more difficulty with evaluating and comprehending complex medical information -including the numerical aspects- were thought to benefit more from visual presentation formats designed to reduce cognitive effort/to increase intuitive/affective meaning of that information. As such, we wanted to focus on people who differ in those (perceived) evaluation/comprehension skills and not so much on (perceived) abilities in information seeking and finding. We agree that the sentence in line 39-40 was too speculative. We have clarified this in the manuscript.

- Lines 39-51. ‘However, understanding medical probability information is greatly influenced by patient skills related to information processing, such as the ability to derive meaning from abstract information and factual evaluation and appraisal of information [5]. Such information processing skills are captured in concepts such as health literacy (HL), which is mainly about accessing, understanding, and using health information [6]; numeracy, which is about understanding and using numbers and probabilities; and graph literacy (GL), which concerns the ability to understand graphically presented information [7]. Concerning HL there is a variety of conceptualizations. While some conceptualizations focus on a broad and holistic view of HL that also includes, for example, searching for information or assessing the reliability of information [e.g., 8], other conceptualizations focus more on specific aspects of HL, such as the literacy aspect [e.g., 9]. In this study, the focus is on understanding and interpreting abstract health information. Therefore HL is conceptualized as an individual ability in which understanding and interpreting health information is central.

2) Your descriptive statistics even with a higher powered sample let nobody expect to find medium effects on comprehension depending on health literacy. Provided an argumentation for an interaction with health literacy, this leads to discuss

a. the stimuli: The icon array and the bar chart in Experiment 1 have a small but relevant error how visualisations are labelled: The legend labels Hormone therapy and Chemotherapy in option 2 and 3 although correctly would be Surgery and Hormone Therapy and Surgery and Chemotherapy, respectively. A scientist may perceive this difference trivial, but actually this misspecification can create confusion in laypeople who aim to understand the legend, particularly if they are not really aware about that they might have already undergone surgery. Items referring to Hormone and Chemotherapy (verbatim!) could bring the visualisation formats at disadvantage. On the other hand, it would be sufficient to read (in Experiment 1) the statement “Alive after 10 years: XX%” – this allows for correct responding to any item and this is constant across presentation conditions. So, why to expect any difference? One could say, the graphs did not mislead them.

RESPONSE: We thank the editor for this attentiveness. We have tried to develop the most optimal visualisations based on co-creation, user testing, and existing guidelines on probability communication. For example, the co-creation sessions revealed that it is important for women that it is immediately clear what the effect of an additional treatment is. Therefore, the legend only mentions this additional effect, something the women in the co-creation sessions and user tests seemed to understand. An aspect also mentioned in the guidelines for communicating probability information is that it must be ensured that people have to make as few mental calculations as possible. This has resulted in the sentence 'alive after 10 years: XX' being included above the visualisations. However, because the survey questions focused on understanding the core information, the questions may not have been best suited to detect differences between the formats. We have now added this to the discussion.

- Lines 440 – 443 ‘It is also possible that despite developing the visualisations in co-creation with the target group, the visualisations were still not optimal. For example, women who were particularly focused on the visualisation and the legend may not have noticed that all the options were about someone who had already undergone surgery.’

- Lines 496 – 500 ‘It should be noted that the comprehension questions were mainly aimed at understanding the core message of the information. This may have resulted in the comprehension questions in Experiment 1 in particular being too easy and not necessarily the most focused on discovering differences between the formats. Therefore, the comprehension questions themselves may also have contributed to the lack of effects.’

b. the participants: Have they just not taken deep notice of the material and answered the questions correctly anyway? [e.g., the gist comprehension questions] Where attention checks for compliant responding included [only three for “poor data quality”, e.g. straightliners, excluded?]? I comprehend the power calculation, but the cells with low literates in the end are not all sufficiently powered, even assuming highly compliant participants.

RESPONSE: We agree that in online experiments such as ours, whether or not respondents complete the questions attentively can play a role. As mentioned in lines 280-281 the panel did examine the data for potential inattentive responders and poor data quality. The panel performed quality checks on open answers, consistency of answers, straight-lining (i.e., the same answer option is chosen throughout a series of statements), and completion time. However, it is possible that participants paid less attention to the information due to the hypothetical scenario. We have therefore added this to the limitations section. 

Regarding the group size of the lower health literate, this is indeed a limitation of the study. We also acknowledge this is our strengths and limitations section in lines 538-543. 

- Lines 523-524 ‘It may be that respondents paid less attention to the information due to the hypothetical scenario.’

c. the measurement. You did pretesting with 30 low-literate participants, but what has been learned? Items with more than 90% correctness in their group did not enter the main survey, anything else on discriminability? Probably the three gist comprehension tasks were too easy for guessing people – many people would expect more treatment more benefit. What are the internal consistencies of comprehension gist and comprehension verbatim and comprehension combined?

RESPONSE: We indeed pretested the questions of the first experiment among 67 women of whom 30 were low health literate. We looked at the number of correct answers and in addition, after each set of questions, respondents had the opportunity to comment on the previous questions. For example, they could indicate that a question was not clear or that there were too many questions. For example, there were six gist questions in the pre-test, but some respondents wondered whether they were real questions or whether they were trick questions because there were so many items that tried to capture more or less the same. That was an extra reason to remove the three gist questions that scored high, i.e., ≥90% correct. Regarding the verbatim questions, there were initially also six questions, but respondents commented that it felt like a math exam. That was also an extra reason to remove the two questions that scored high, i.e., ≥90% correct. Respondents also thought that the questionnaire as a whole was too long and that is why we omitted some questions. For example, we reduced the number of questions for the perception of treatment effect from six to two and only included the questions that were about the comparison of hormone treatment versus no additional treatment and the comparison of hormone treatment versus hormone treatment and chemotherapy. Furthermore, minor textual adjustments were made based on the open comments. We have now explained this more clearly in the manuscript and we have added to the discussion that the comprehension questions, particularly in the first experiment, may have been too easy and may not have been sufficiently focused on finding differences between the formats. 

The internal consistencies of the comprehension scales are added to Table 3. No combined score for the gist and verbatim comprehension has been calculated. As outlined in the introduction, gist and verbatim comprehension are two different concepts that can have different effects on the understanding of visualisations.

- Lines 236-241 ‘To avoid ceiling effects in the experiment and to take into account respondents' comments that the number of questions made it feel like a math exam, we selected comprehension questions that were answered correctly by ≤90% of women with lower HL. In addition, the number of questions was reduced because respondents indicated that the questionnaire was too long and minor textual adjustments were made if respondents indicated that the question or response category was not clear.’

- Lines 496-500 ‘It should be noted that the comprehension questions were mainly aimed at understanding the core message of the information. This may have resulted in the comprehension questions in Experiment 1 in particular being too easy and not necessarily the most focused on discovering differences between the formats. Therefore, the comprehension questions themselves may also have contributed to the lack of effects.’

- Lines 246-248 ‘The Kuder-Richardson Reliability Coefficient for Experiment 1 is .71 for gist comprehension and .57 for verbatim comprehension. For Experiment 2 the Kuder-Richardson Reliability Coefficient is .44 for gist comprehension trade-off and .67 for gist comprehension side-effects probability.’

d. the analysis. Have you considered format analysis across all items (simple comprehension sum score)? Have you considered a sensitivity analysis excluding those, who respond “I don’t know”?

RESPONSE: As also indicated in the answer above, we assume that gist comprehension and verbatim comprehension measure different aspects of comprehension. That is why we did not want to calculate a total score. 

We added the answer category 'I don't know' to prevent people who did not know the answer from filling in a random answer. It was mandatory to fill in an answer, so that could lead to respondents guessing the answer. Because the comprehension questions are about measuring understanding, we believe that respondents who answered 'I don't know' should not be excluded.

3) Please refer not only to shared decision making but to the goal of health communication enabling informed decisions according to evidence-based medicine, Western health system standard. Particularly, informing patients about benefits and harms is one of many rules according to established guidelines on how to design health communication (e.g., .

RESPONSE: We agree with the editor that this elaboration is useful. Accordingly, we have made adjustments in both the abstract, introduction, and discussion.

- Lines 3-4 ‘If communicated adequately, numerical decision-relevant information can support informed and shared decision-making.’

- Lines 32-34 ‘Informing patients about benefits and harms of different options is one of the key principles in health communication regarding informed and shared decision making (SDM) [1, 2].’

- Lines 424-426 ‘In this study, we investigated the effects of several (visual) presentation formats to present decision-relevant numerical information (i.e., survival rates and side-effects) to patients in support of informed and shared decision making (SDM).’

4) Also, how did you arrive at subscale of decisional conflict (instead of the full assessment?) Could you derive in the introduction why it is relevant to assess how someone could have felt informed?

RESPONSE: The Decisional Conflict scale consists of five subscales. In the second experiment, we used the Informed subscale as a primary outcome measure and the Uncertainty subscale as a secondary outcome measure. We have not included the other subscales because they concern value clarity, support from others, and the effectiveness of the decision. Because our experiment did not include explicit value clarification exercises nor was it presented in a context in which support from others could have played a role, these subscales did not seem relevant to our study. Instead, we used the six items of the preparedness for decision-making scale which assess the extent to which the information has been supportive. The rationale was that the presentation of decision-relevant information could help people feel more prepared for decision-making. 

The secondary outcome measures such as feeling informed and the evaluation of the information were included because we know that decision-relevant information can have a different influence on such measures compared to the influence on comprehension (Trevena et al., 2021). We have added the following to the introduction: 

- Lines 119-123 ‘In addition, we also assessed the perception of the treatment effect in the first experiment and risk perception regarding additional treatment in the second experiment. Because the information presented was intended to support decision-making, we also included decision uncertainty and the extent to which the information contributes to the perceived preparedness for decision-making in the second experiment.’

5) Abstract

a. “When communicated adequately..”

b. Capitalisation of shared decision making and health literacy seems unusual

c. When high/low …. Perhaps better expressed “depending on their”….

d. Probability information in numbers/visualisations … Perhaps better expressed “numbers with or without…” Numbers accompanied also visualisations here.

RESPONSE: We thank the editor for his careful reading. We have made changes to the abstract and where appropriate to the rest of the manuscript, e.g., omitting the capitalization. 

- Lines 3-5. ‘If communicated adequately, numerical decision-relevant information can support informed and shared decision making. Visual formats are recommended, but which format supports patients depending on their health literacy (HL) levels for specific decisions is unclear.’

- Lines 98-100 ‘In addition, previous research has not investigated which format best suits people depending on their information processing skills.’

- Lines 13-17. ‘Experiment 2 had a 5 (side-effects format: no probability information – probability information in numbers with or without a visualisation – probability information in numbers with or without a visualisation accompanied by a description of the side-effects) x 2 (HL: low – high) design.’

6) Introduction; generally: the impression of specific visualisation that outperforms no visualisation should be avoided, because state of evidence is that different presentation formats are beneficial for different problems and different dialog groups. Please leave it in a format comparison, as you analysed it, bar vs. text and

---

## [Editor Report · Decision Letter 2]

26 Jun 2024

PONE-D-23-32983R2Presenting decision-relevant numerical information to Dutch women aged 50-70 with varying levels of health literacy: case example of adjuvant systemic therapy for breast cancerPLOS ONE

Dear Dr. van Strien,

Thank you for submitting your manuscript to PLOS ONE. After careful consideration, we feel that it has merit but does not fully meet PLOS ONE’s publication criteria as it currently stands. Therefore, we invite you to submit a revised version of the manuscript that addresses the points raised during the review process.

We look forward to receiving your revised manuscript.

Kind regards,

Felix G. Rebitschek

Academic Editor

PLOS ONE

Journal Requirements:

Additional Editor Comments:

Dear Authors,

You have extensively addressed concerns and corrected where necessary. Thank you! Finally, one thing remains, which centers around low-literacy findings/non-findings of Experiment 2.

In your preregistration you hypothesised:

H2a. People provided with information in a visualization [..] will report more adequate comprehension of the trade-off between survival rates and likelihood of side-effects (gist comprehension trade-off) compared to people provided with the information in numbers only (condition 2a and 3a). Health literacy will moderate this relation, in the sense that those with lower/inadequate health literacy will be better supported in comprehension of the trade-off with the visualization (condition 2b – condition 3b) compared to numbers only (condition 2a and 3a) compared to those with higher/adequate health literacy. Similarly, H2b.

Then you tested that according to the manuscript:

"in both experiments the analyses with comprehension as outcome were performed with cumulative odds ordinal logistic regression with proportional odds (instead of ANOVAs)."

The power calcuation, however, was made for an ANOVA. And you in the discussion: "However, it should be noted that the expected interaction-effects were ordinal-interactions rather than full crossover interactions, therefore the statistical power to detect the expected interactions is lower than the a priori calculated 90% and 91%."

Now, Experiment 2 that varies the presentation of probability came with cell samples between 17 and 26 participants. You referred to both the power calculation (which was made under different assumptions) and limitation section in your discussion as cited.

Now we can say that Experiment2-Comprehension was less likely to detect any difference among the factor stages (than planned), but even less likely to enable planned comparisons as you hypothesised them in the preregistration.

Btw: With which tool have you done the power calculation?

On the other side you wrote about applying Bonferroni correction but your results seem to be interpreted still consistently under alpha<.05. How do I recognise your adjustment? For the example: “Regarding ‘feeling informed’ (H3), we found an interaction between HL and format, F(4, 274) = 2.67, p = .032, partial η2 = .04.“ If alpha is adjusted, this may typically would not count as being significant. Note that according to the preregistration the interaction on „feeling informed“ is exploratory.

Taken together, please check how both low power and multiple testing correspond with your results and discussion on the interaction with HL; according to my understanding the manuscript would benefit if you make more concrete that the question whether low HL people comprehended your interventions differently than high HL people could not be reliably addressed by your studies.

---

## [Author Response · Author response to Decision Letter 2]

18 Jul 2024

Dear Editor,

Thank you for the comments and the opportunity to address these points in our manuscript entitled ‘Presenting decision-relevant numerical information to Dutch women aged 50-70 with varying levels of health literacy: case example of adjuvant systemic therapy for breast cancer’ (PONE-D-23-32983R2). We have responded to the outstanding points as described below. Changes to the manuscript are underlined in our responses and indicated by tracked changes in the revised manuscript. 

Additional Editor Comments:

Dear Authors,

You have extensively addressed concerns and corrected where necessary. Thank you! Finally, one thing remains, which centers around low-literacy findings/non-findings of Experiment 2.

In your preregistration you hypothesised:

H2a. People provided with information in a visualization [..] will report more adequate comprehension of the trade-off between survival rates and likelihood of side-effects (gist comprehension trade-off) compared to people provided with the information in numbers only (condition 2a and 3a). Health literacy will moderate this relation, in the sense that those with lower/inadequate health literacy will be better supported in comprehension of the trade-off with the visualization (condition 2b – condition 3b) compared to numbers only (condition 2a and 3a) compared to those with higher/adequate health literacy. Similarly, H2b.

Then you tested that according to the manuscript:

"in both experiments the analyses with comprehension as outcome were performed with cumulative odds ordinal logistic regression with proportional odds (instead of ANOVAs)."

The power calcuation, however, was made for an ANOVA. And you in the discussion: "However, it should be noted that the expected interaction-effects were ordinal-interactions rather than full crossover interactions, therefore the statistical power to detect the expected interactions is lower than the a priori calculated 90% and 91%."

Now, Experiment 2 that varies the presentation of probability came with cell samples between 17 and 26 participants. You referred to both the power calculation (which was made under different assumptions) and limitation section in your discussion as cited.

Now we can say that Experiment2-Comprehension was less likely to detect any difference among the factor stages (than planned), but even less likely to enable planned comparisons as you hypothesised them in the preregistration.

Btw: With which tool have you done the power calculation?

On the other side you wrote about applying Bonferroni correction but your results seem to be interpreted still consistently under alpha<.05. How do I recognise your adjustment? For the example: “Regarding ‘feeling informed’ (H3), we found an interaction between HL and format, F(4, 274) = 2.67, p = .032, partial η2 = .04.“ If alpha is adjusted, this may typically would not count as being significant. Note that according to the preregistration the interaction on „feeling informed“ is exploratory.

Taken together, please check how both low power and multiple testing correspond with your results and discussion on the interaction with HL; according to my understanding the manuscript would benefit if you make more concrete that the question whether low HL people comprehended your interventions differently than high HL people could not be reliably addressed by your studies.

RESPONSE: We thank the editor for his comments regarding low power and multiple testing in the second experiment. We agree with the editor that our study has limitations in answering the question of whether people with low health literacy were better supported in understanding the side-effects when visualizations were used than people with high health literacy. We have now stated this more clearly in the Discussion section. 

As mentioned in the pre-registration, we used the software programs G*Power and PASS for the power calculations. We have now also added this to the manuscript in the Methods section. 

Regarding the Bonferroni correction, this adjustment has been incorporated into simple main effects results. We have now also indicated this in the manuscript in the Results section. We hope that this makes it more recognizable in the manuscript. 

- Lines 293-294 ‘An a priori power analysis was performed based on a 3x2 (Experiment 1) and a 5x2 factorial ANOVA design (Experiment 2) with interaction-effect using the software programs G-power and PASS.’

- Lines 383-385 ‘For format C (visualised probability information without description), there was a difference in the average score on feeling informed between women with low and high HL, F (1,272) = 7.75, p = .006 after Bonferroni correction, partial η2 = .03.’

- Lines 535-537 ‘This implies that the question of whether people with lower health literacy levels would benefit more from the formats with the visualizations than people with higher health literacy could not be answered reliably.’

---

## [Editor Report · Decision Letter 3]

16 Aug 2024

Presenting decision-relevant numerical information to Dutch women aged 50-70 with varying levels of health literacy: case example of adjuvant systemic therapy for breast cancer

PONE-D-23-32983R3

Dear Dr. van Strien,

We’re pleased to inform you that your manuscript has been judged scientifically suitable for publication and will be formally accepted for publication once it meets all outstanding technical requirements.

Kind regards,

Felix G. Rebitschek

Academic Editor

PLOS ONE
---

## [Editor Report · Acceptance letter]

22 Aug 2024

PONE-D-23-32983R3 

PLOS ONE

Dear Dr. van Strien-Knippenberg, 

I'm pleased to inform you that your manuscript has been deemed suitable for publication in PLOS ONE. Congratulations! Your manuscript is now being handed over to our production team.

Kind regards, 

on behalf of

Dr. Felix G. Rebitschek 

Academic Editor

PLOS ONE